# Recognition of Cursive Pashto Optical Digits and Characters with Trio Deep Learning Neural Network Models

**Muhammad Zubair Rehman [1,2], Nazri Mohd. Nawi [1], Mohammad Arshad [3] and Abdullah Khan [3,\*]**

1   Soft Computing and Data Mining Centre (SMC), Faculty of Computer Science & Information Technology, University Tun Hussein Onn Malaysia (UTHM), Batu Pahat 86400, Malaysia; syedzubair@uthm.edu.my (M.Z.R.); nazri@uthm.edu.my (N.M.N.)

2   Faculty of Computing and IT, Sohar University, Sohar 311, Oman

3   Faculty of Management and Computer Sciences, Institute of Computer Sciences and Information Technology, The University of Agriculture, Peshawar 25120, Pakistan; mohammadarshad.444@gmail.com

\*   Correspondence: abdullah_khan@aup.edu.pk

**Abstract:** Pashto is one of the most ancient and historical languages in the world and is spoken in Pakistan and Afghanistan. Various languages like Urdu, English, Chinese, and Japanese have OCR applications, but very little work has been conducted on the Pashto language in this perspective. It becomes more difficult for OCR applications to recognize handwritten characters and digits, because handwriting is influenced by the writer's hand dynamics. Moreover, there was no publicly available dataset for handwritten Pashto digits before this study. Due to this, there was no work performed on the recognition of Pashto handwritten digits and characters combined. To achieve this objective, a dataset of Pashto handwritten digits consisting of 60,000 images was created. The trio deep learning Convolutional Neural Network, i.e., CNN, LeNet, and Deep CNN were trained and tested with both Pashto handwritten characters and digits datasets. From the simulations, the Deep CNN achieved 99.42 percent accuracy for Pashto handwritten digits, 99.17 percent accuracy for handwritten characters, and 70.65 percent accuracy for combined digits and characters. Similarly, LeNet and CNN models achieved slightly less accuracies (LeNet; 98.82, 99.15, and 69.82 percent and CNN; 98.30, 98.74, and 66.53 percent) for Pashto handwritten digits, Pashto characters, and the combined Pashto digits and characters recognition datasets, respectively. Based on these results, the Deep CNN model is the best model in terms of accuracy and loss as compared to the other two models.

**Keywords:** OCR; Pashto digits; convolutional neural networks; Deep CNN; LeNet

## 1. Introduction

Having a basic knowledge of the reading and semantics of any specific language or script enables a native human to easily read and understand text documents in that language. The same task is performed through Optical Character Recognition (OCR), which converts handwritten or printed scripts into machine language [1]. Developed by Germany and later on used extensively in the US, OCR utilizes computer vision, which enables a machine to recognize different characters of any language [2]. OCR plays an important role in the digitization of text documents. Digitization enables a document's content to be searched on the web, making it compatible with other digital applications, and reduces the cost of physical storage. OCR is one of the branches of document image analysis (DIA), which itself is one of the major branches of pattern recognition. Significant research has been conducted in the past decade [3] in the field of OCR [4]. Different languages have their own OCR systems such as Chinese [5], German [6], Japanese [7], English [8], Indic based script [9], and French [10]. However, languages that have cursive scripts such as Pashto [11,12], Arabic [13], Persian [14], Sindhi [15], and Urdu [16] still require further work. The demand to translate and understand different regional languages on the Asian

subcontinent has increased tenfold in this era. Pashto is written and spoken by 50 million people in the world and has a rich literacy history. However, a very limited amount of work has been conducted to digitize Pashto handwriting in the recent age. Pashto has a linguistics composition and character shapes very similar to Arabic. This quality of being specific needs devoted research to find a conceptual base for the challenges set by the Pashto language in the field of OCR. Although less work has been conducted for Pashto OCR, still no work has been conducted for Pashto handwritten digits' recognition, and therefore there is no publicly available dataset for Pashto digits. A dataset for Pashto handwritten characters was produced by [17]. Similarly, a dataset of "Printed sentence" was produced by [16] for their study, but a handwritten digits dataset has not yet been produced. Due to the similarity in writing styles, the printed characters or digits are easy to train and recognize [18], but handwritten scripts vary from person to person, which is somehow easy for humans to understand but becomes a challenging task for a machine to recognize, especially when there are multiple shapes for a single character. To overcome this challenge of training a machine with Pashto digits, a proper dataset for Pashto digits is needed. Thus, a proper dataset for Pashto digits is one of the basic motivations behind this study. This study proposes the use of Deep Convolutional Neural Network (DCNN) for recognition of three different datasets, i.e., Pashto digits, Pashto characters, and combined Pashto digit and character datasets.

The main contributions of this study are:

1. To develop a Benchmark Handwritten Pashto digits dataset.
2. To build a Deep Convolutional Neural Network having seven sets of Convolution, ReLU, and Maxpooling function for the classification of Pashto digits, Pashto characters, and combined Pashto digit and characters datasets.
3. To check the performance of the trio deep learning Convolutional Neural Network, i.e., CNN, LeNet, and DCNN in terms of parameters such as accuracy, loss, precision, recall, and f-measure.

The next section discusses the previous related work on OCR. Section 3 elaborates on the proposed methodology for the Pashto OCR used in this paper. Section 4 presents the simulations' results, and finally the paper is concluded in Section 5.

## 2. Related Work

This section will briefly explain the previous work conducted on the techniques that have been applied by different researchers in the field of OCR using different cursive languages' printed or handwritten text. A larger challenge these days is to recognize different cursive languages such as Urdu, Pashto, Arabic, Persian, Chinese, and Latin [13]. As far as Pashto is concerned, currently no work has been conducted to recognize Pashto handwritten digits, while for Pashto handwritten characters only two reputed works have been completed [19].

One of the early works on Pashto was completed by using Hidden Markov Models (HMM). An OCR system was implemented for BBN Byblos (Beranek, Bolt and Newman technologies) and the machine was trained to recognize Pashto documents containing printed data. The simulation results showed an error rate of 1.6 percent for synthetic images, a 2.1 percent error rate for scanned pages, and the error was 3.1 percent for faxed pages [20]. In another study, Principal Component analysis and Scale Invariant Feature Transformation (SIFT) were used to overcome the challenges faced while working with cursive scripts such as Pashto, Arabic, and Urdu. The approaches such as recurrent neural network and Long Short-Term Memory (LSTM) were good and an accuracy of 89–94 percent was achieved in the text recognition [21].

Another study was conducted on real world-based Urdu and Pashto text data. In this study, a written text which does not remain straight but rotates in any direction was detected. Different OCR techniques such as Scale Invariant Feature Transformation (SIFT), Hidden Markov Model (HMM), and Long Short-Term Memory to recognize rotated text were used. An accuracy of 98.9 percent was obtained by LSTM, 94.3 percent by SIFT, while

HMM showed an accuracy of 89.9 percent [19]. In a more recent study, a recurrent neural network was examined for non-cursive and cursive scripts by using the Bidirectional Long Short-Term Memory (BLSTM) model. BLSTM is a variant of the Recurrent Neural Network with a special layer called connectionist temporal classification (CTC). The BLSTM network achieved an accuracy of 98.75 percent in cursive script recognition [22]. In a similar study on Pashto, a dataset consisting of 1000 Pashto unique ligatures was created [23]. This study was later on extended to create OCR systems in 2018 [16]. Like Pashto, Urdu text was classified at a ligature level in this research. Character level segmentation problems associated with cursive language scripts were overcome by the use of machine learning techniques such as Naive Bayes, discriminant analysis, and decision tree. An accuracy of 73, 61, and 62 percent was claimed by Naive Bayes, discriminant analysis, and decision tree, respectively [24].

Similarly, an Optical Character Recognition System was proposed that used Arabic, Urdu, and Pashto in such a way that the similar properties of one language could be tested in another language by using the printed text of these languages. The effect of different languages on recognition accuracy was investigated when they were combined using synthetic datasets that are publicly available for the Pashto and Arabic languages. The statistical analysis was also provided as clues for transfer learning concerning OCR systems for Pashto, Arabic, and Urdu languages. A dataset named KPTI of Pashto gave 25 percent accuracy, and a UPTI dataset for the Urdu language showed an accuracy of 38 percent after training MDLSTM on an Arabic dataset KHAT; it was tested on unseen data [25].

An OCR system was developed that used the Feed Forward Neural Network (FFNN) with each input layer consisting of 315 neurons, with 2000 neurons in a hidden layer using $21 \times 15$ pixel symbols, yielding a result in a six node output layer for the recognition of joinable printed Pashto characters on different locations in scanned or printed Pashto continuous text and achieved an accuracy of 78 percent [26].

In a more recent work on Pashto, an OCR system was produced at the University of Kaiserslautern, Germany for Pashto cursive script in 2018, which was also used for recognizing different Pashto books' text using various deep learning techniques [16]. The MDLSTM model produced a 9.22 percent error rate, and a 16.16 percent character error rate was detected for BLSTM. In the same year, another study used their custom dataset and applied a zoning technique with KNN and ANN. Their dataset consisted of 4488 images with 44 classes with each character having 102 images. This study achieved 72 percent and 70.02 percent accuracy on ANN and KNN, respectively.

The literature shows that most of the work is completed in OCR systems for various languages (English, Latin, Persian, Arabic, Urdu, and Pashto) on printed scripts and little work is completed for handwritten script. Due to the changes in the writing style of each individual, OCR detection becomes complex; on the other hand, printed text does not pose problems due to being less complex. There is a dataset available for Pashto handwritten characters, but no dataset is available for Pashto digits and hence no work has been conducted previously on Pashto digits [13,23]. Therefore, this research focused on Pashto handwritten character as well as digits recognition with a deep convolutional neural network. The dataset for Pashto characters was obtained from GitHub [17], and for Pashto digits the dataset was created in this study.

## 3. The Proposed Methodology

The proposed research process will be discussed in this section. Figure 1 shows the overall research methodology. The substeps involved in the research process are discussed in detail in the rest of the section. First, the basis of the motivation of this study is explained; the approach to data collection is discussed in the next section, while section A explains different preprocessing techniques that were applied to the dataset. Further, the selected variables for the experiment will be discussed. Similarly, the next section describes the network design, while the final section sheds light on the validation of the model.

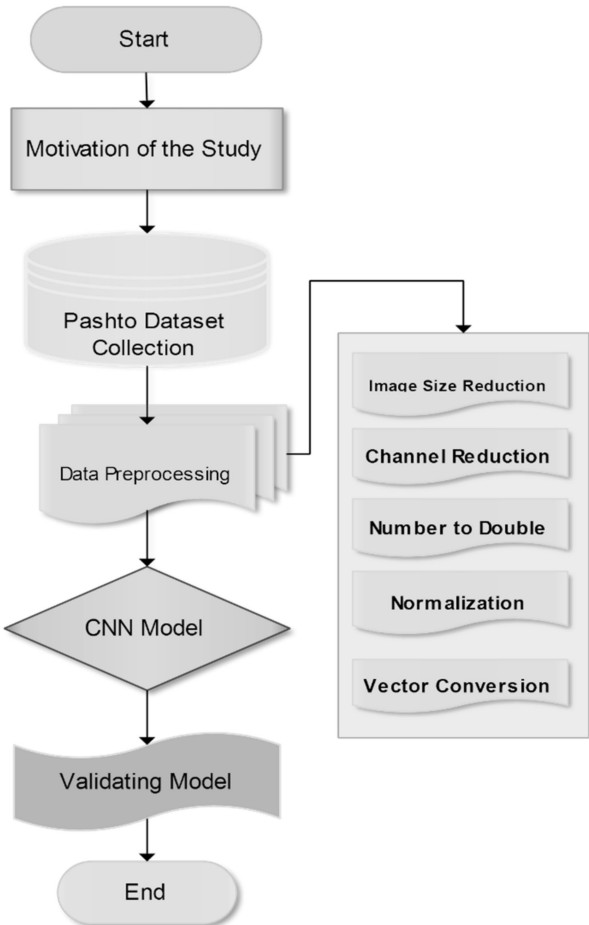

**Figure 1.** The Proposed Methodology Flowchart.

All the stages shown in Figure 2 are explained in the subsections. After the collection of the data, each table was split into 50 pieces so that each digit could be extracted separately as shown in Figure 3. A canny edge detection function was used for separating each cell. A total of 27,500 images were formed, after dividing 550 images from 50 images. Then each image was passed through a neural network.

### 3.1. Data Collections

The Pashto character dataset was downloaded from GitHub [17]. It was created in a previous study by the University of Agriculture, Peshawar. A total of 43,000 sample images were collected from the university students. Pashto characters are given in Table 1. A Pashto digits dataset was created in this study by collecting handwritten digit samples from a total of 550 students at school level. A total of 60,000 digit images were divided in a ratio of 80 percent training and 20 percent testing data following the baseline sample as given in Figure 4. The Pashto digits and characters datasets were combined to form an image dataset. The dataset consisting of 106,000 for 53 classes (43 for characters and 10 for digits) was divided into 80 percent for training and 20 percent for testing.

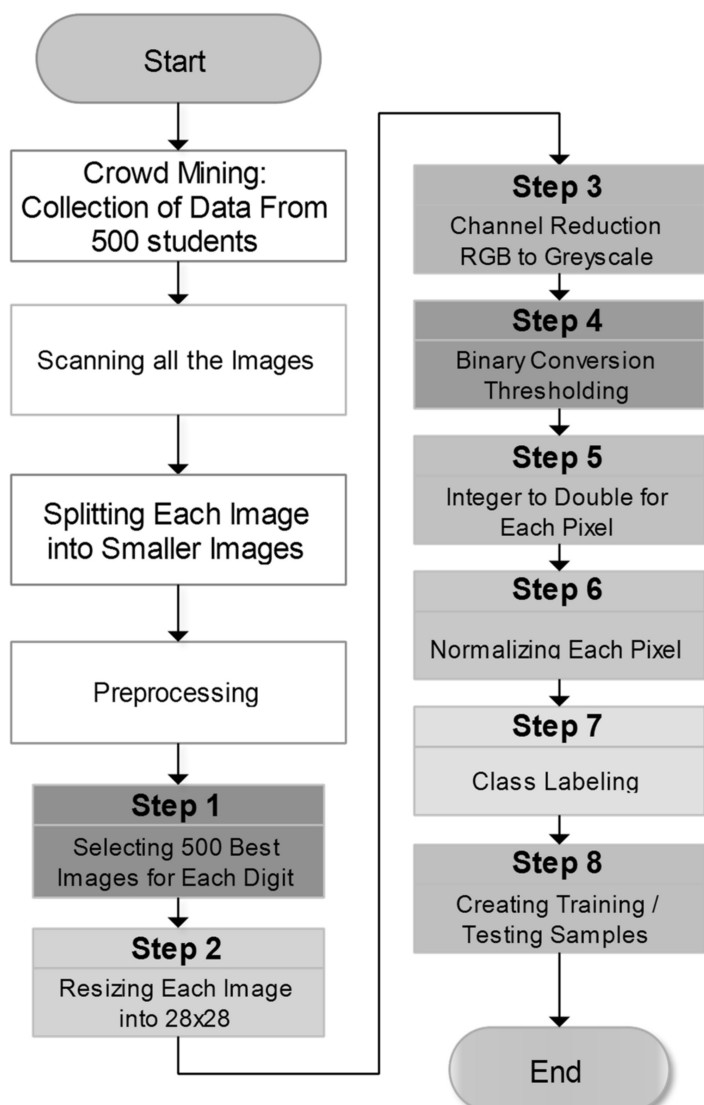

**Figure 2.** The Pashto Digit Dataset Collection and Preprocessing.

**Figure 3.** A sample from the Pashto Digit Dataset.

**Table 1.** Pashto Characters Dataset.

| خ | ح | چ | ج | ث | ټ | ت | پ | ب | ا |
|---|---|---|---|---|---|---|---|---|---|
| ژ | ز | ړ | ر | د | ذ | ډ | د | ځ | څ |
| غ | ع | ظ | ط | ض | ص | ښ | ک | س | ق |
| ه | و | ڼ | ن | م | ل | ګ | ک | ق | ي |
|   |   |   | ئ | ی | ی | ي | ی | ي | ۀ |

| ٩ | ٨ | ٧ | ٦ | ٥ | ٤ | ٣ | ٢ | ١ | ٠ |
|---|---|---|---|---|---|---|---|---|---|
| نهه | اته | اووه | شپږ | پنځه | څلور | درے | دوه | يو | صفر |

**Figure 4.** Complete Pashto Digits.

### 3.2. Data Preprocessing

After the dataset collection, it was preprocessed to remove noise from the dataset and to filter it. Open CV and Python tools were used for preprocessing. The data were preprocessed in the following steps. Some of the images were not able to be entered into the CNN for machine learning, therefore those images were dropped. The discarded images are shown in Figure 5.

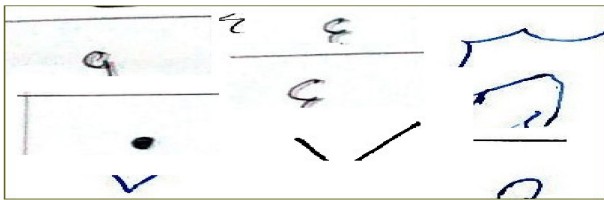

**Figure 5.** Discarded images.

Similarly, the image size was decreased to bring it to a stable size which was able to be entered into CNN; the larger the image size, the more time it takes to be processed. The delay can be decreased by dimensionality reduction. The standard set by MNIST dataset was followed to reduce the pixel size of each image's rows and columns to 28 × 28 making a standard input of 784 pixels. The images were scanned having three-channel RGB, which were converted from a 3-channel color (RGB) into a single channel grayscale image. The channels were reduced by using an Open CV function; the image in RGB was a three-channel image of 28 × 28 × 3 as shown in Figure 6 while the actual conversion is shown in the Figure 7.

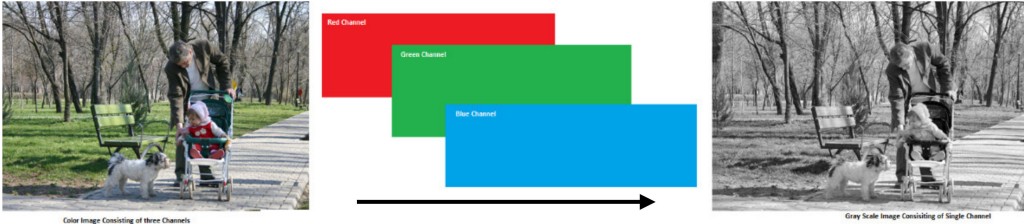

**Figure 6.** Channel Representation of RGB.

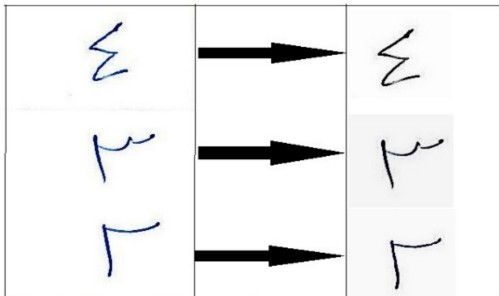

**Figure 7.** Actual conversion of characters.

The grayscale images were converted into binary images. The shades of grayscale images are between gray and white; on the other hand, binary images have only black and white pixels. An image can be converted into binary by making all the pixels above a certain specific threshold value into black, while the remaining is kept white. The binary digit images can be seen in Figure 8.

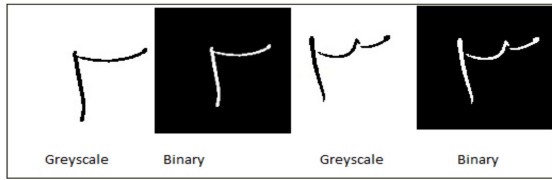

**Figure 8.** Grayscale vs. binary.

The values of the image matrix that were between 0 and 255 were converted to double precision values because the convolutional neural network performs better when input values are normalized. The image is actually a matrix of numbers as shown in Figure 9.

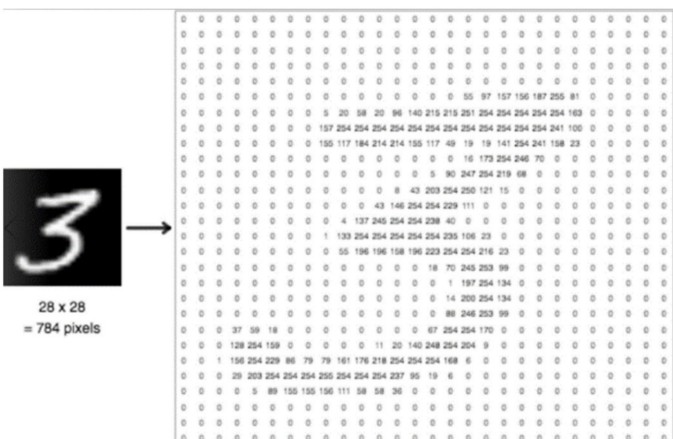

**Figure 9.** Image representation.

The NumPy "astype" method was used to convert pixel values into double. This method converted each image into a double precision number. The images' pixel values were normalized using a min–max normalized method. After achieving the minimum and maximum values of the pixels, each image was normalized by traversing its pixel value using Equation (1).

$$\text{Pixel}_{\text{img}} = \frac{(\text{Pixel}_{\text{img}} - \min)}{(\max - \min)} \tag{1}$$

where max and min are the maximum and minimum values of any pixel. The pixel values were normalized between 0 and 1 but the actual intensity of the image was not disturbed. Humans cannot write like a computer in a fixed shape, size, and orientation. Image

augmentation is a technique that rotates the writing in an image with certain angles and produces a new variety of images. In this way, a dataset can be made larger, and hence prevents overtraining. Python libraries, such as Pillow (PIL Fork 8.3.1) and OpenCV for Python 4.5.3.56, were used to apply image augmentation. A total of 60,000 handwritten digits images were produced having 6000 images for each digit by rotating random images by 20 degrees in random directions. The images for each digit were moved to corresponding folders and then that folder name was set as a label for that particular digit.

**Data Partitioning:** The whole dataset could not be used in training. The dataset were split between testing and training, then training data was be used to train the model and testing data was used to predict the unseen data. In this study, training and testing data were split into 80 and 20 percent, respectively.

**Deep Neural Network Models:** This paper uses the Adagrad optimization function in CNN, LeNet, and Deep CNN models, with each model having different sets of convolutions, max pooling, and ReLU activation layers. The deep convolutional neural network provided the ability to train and gave close to real world approximations. The final fully connected layer in CNN used the SoftMax function to calculate the probabilities of the predicted values in a multiclass classification problem. So, the importance of the SoftMax activation function in the final layer cannot be ruled out. In simple CNN, only one set of convolutional, max pooling, and ReLU activation layers were used with only two layers fully connected and the final layer as the SoftMax classifier. In LeNet, two sets of convolutional, max pooling, and ReLU activation layers were used with only two layers fully connected and the SoftMax classifier as the last layer. In the Deep CNN model, four sets of convolutional, ReLU activation, batch normalization, max pooling, and dropout layers were used having two layers fully connected and the final layer as the SoftMax classifier.

*3.3. Performance Parameters*

To evaluate the performance of the proposed CNN, LeNet, and Deep CNN models in this research, different performance parameters were used.

$$\text{Accuracy} = \frac{\text{TP} + \text{TN}}{\text{TP} + \text{TN} + \text{FP} + \text{FN}} \tag{2}$$

$$\text{Precision} = \frac{\text{TP}}{\text{FP} + \text{TP}} \tag{3}$$

$$\text{Recall} = \frac{\text{TP}}{\text{TP} + \text{FN}} \tag{4}$$

$$\text{F}_1 - \text{Score} = 2.\frac{\text{precision . recall}}{\text{precision} + \text{recall}} \tag{5}$$

$$\text{MSE} = \frac{1}{n} \sum_{i=1}^{n} \left( X_i - X_i' \right)^2 \tag{6}$$

## 4. Results and Discussion

The performance of the proposed model was further tested and validated. The proposed model (Deep CNN)'s performance was compared with CNN and LeNet. The proposed Deep CNN had more convolutional layers than the other two models. Pashto characters, Pashto digits datasets, and finally the combined Pashto digits and character dataset were used in this study. Accuracy, loss, precision, F1-score, and recall were used to check the performance of the models. The models were applied both on training and testing data.

For recognition of Pashto characters, the dataset was created in the previous study and is available online on GitHub [17]. A total of 2150 samples were collected from 350 students, which reached up to 43,000 images after preprocessing and augmentation. Each character had 1000 images. For Pashto digits, the dataset was collected in this study. The data was collected from 500 students having 25,000 samples initially and after augmentation the

dataset reached up to 60,000 images. Each digit had 6000 images of that digit. For the combined Pashto digit and character, we used 2000 images for each class. The data was split into a ratio of 80:20 for training and testing purposes.

*4.1. Preliminaries*

Two main evaluation metrics of this study were accuracy and loss. Each dataset was split into 80:20 for training and testing. The experiments were performed on an Intel 2.0 GHz Core i3 processor having 4 GB RAM. The operating system was Windows 10. The Python library Keras was used for training and testing the model for all the datasets. The proposed Deep CNN algorithm was investigated by comparing it with CNN and LeNet. The proposed Deep CNN had seven sets of Convolutional, Relu, and Max Pooling and two sets of fully connected layers. All the experiments had a default learning rate of 0.01. All three algorithms were trained and tested with random biases and weights. All algorithms were trained on a maximum of 20 epochs and used the SoftMax activation function for multiclass classification.

*4.2. Experiments*

Experiments were performed on the Pashto Character, the collected Pashto Digit, and the combined Pashto Digit and character datasets. Three models were applied on every dataset. Accuracy, loss, recall, and F-measure were used to evaluate the models. The progress of each evaluation metric was recorded for training and testing data after every epoch. Each experiment for each dataset will be explained in detail in the upcoming sections.

*4.3. Pashto Character Dataset*

The Pashto character dataset was divided into a ratio of 80:20 for training and testing data. All the models were trained in 20 epochs. Comparison of the evaluation metrics are given in Table 1.

Table 2 shows the accuracy of the proposed model as compared to the sample CNN and LeNet models. The proposed Deep CNN achieved 99.4% accuracy, while the CNN and LeNet model achieved 98.3% and 99.25% accuracy. From the simulation results, the proposed model outperformed the compared models in terms of accuracy for the Pashto character dataset. For the loss comparison of the proposed model as compared to the sample CNN and LeNet, the proposed Deep CNN had a loss of 0.020, while the CNN and LeNet models had a loss of 0.05 and 0.027. From the simulation results, the proposed model outperformed the compared models in term of the loss for the Pashto character dataset. The comparison of accuracy and loss can be seen in Figures 10 and 11.

**Table 2.** Accuracy and Loss Comparison of CNN, LeNet, and Deep CNN Models on the Pashto Character Dataset.

| Accuracy/MSE | CNN | LeNet | Deep CNN |
|:---:|:---:|:---:|:---:|
| Accuracy | 98.3 | 99.25 | 99.4 |
| MSE | 0.057 | 0.027 | 0.020 |

Similarly, the comparison performance of the three models in terms of precision, recall, and F-Score, are shown in Table 3. From Table 3, it can be clearly seen that the CNN model had a precision of 0.983 percent, a recall value of 98.2 percent, and an F1-Score of 98.3 percent. The LeNet had the same value of 99.2 percent for all three parameters, i.e., precision, recall, and F1-Score. The proposed Deep CNN showed better performance in terms of precision, recall and F1-Score; it had a precision value of 99.4 percent, a recall value of 99.4 percent, and an F1-Score of 99.4 percent. Figure 12 shows the performance of these three parameters on all three models.

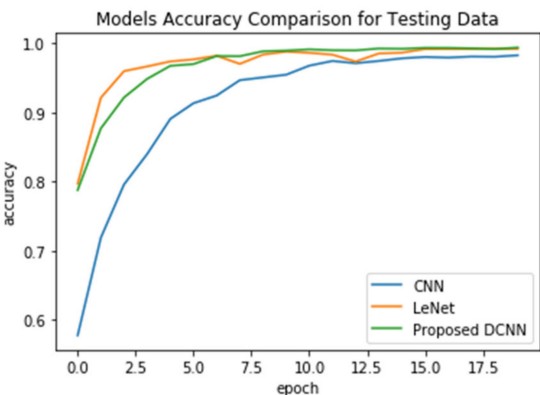

**Figure 10.** Accuracy Comparison of CNN, LeNet, and the Proposed DCNN on the Pashto Characters Dataset.

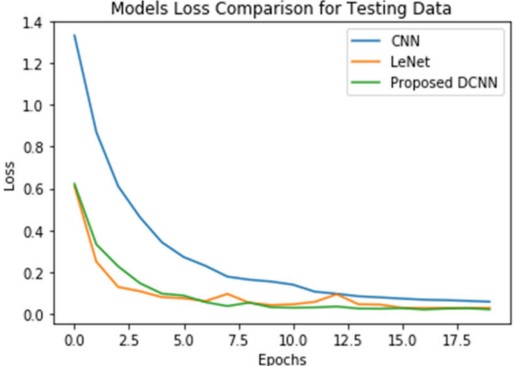

**Figure 11.** Loss Comparison of CNN, LeNet, and the Proposed DCNN on the Pashto Characters Dataset.

**Table 3.** Precision, Recall, and $F_1$-Score Comparison of CNN, LeNet, and Deep CNN Models of the Pashto Character Dataset.

| Parameters | CNN | LeNet | Deep CNN |
|:---:|:---:|:---:|:---:|
| Precision | 98.3 | 99.2 | 99.4 |
| Recall | 98.2 | 99.2 | 99.4 |
| F-Score | 98.3 | 99.2 | 99.4 |

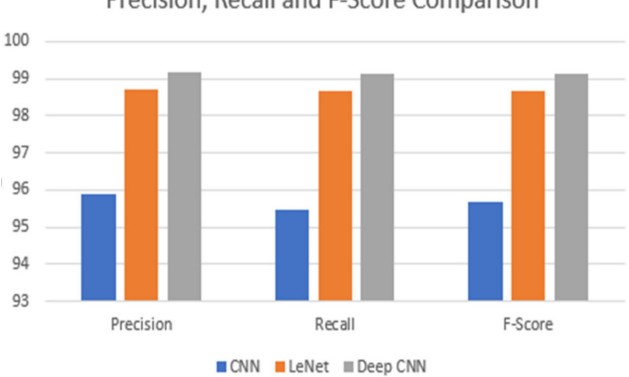

**Figure 12.** Precision, Recall, and $F_1$-Score Comparison of CNN, LeNet, and the Proposed DCNN on the Pashto Characters Dataset.



*4.4. Pashto Digit Dataset*

The Pashto digits dataset was created in this study by collecting handwritten digit samples from a total of 550 students at school level. A total of 60,000 digits images were divided into a ratio of 80 percent training and 20 percent testing data. CNN, LeNet, and Deep CNN models were applied. Table 4 shows the accuracy and loss. Similarly, Table 5 shows the precision, recall, and F1-Score comparisons of these models.

**Table 4.** Accuracy and Loss Comparison of CNN, LeNet, and Deep CNN Models on the Pashto Character Dataset.

| Accuracy/MSE | CNN | LeNet | Deep CNN |
|---|---|---|---|
| Accuracy | 98.7 | 91.9 | 99.1 |
| MSE | 0.031 | 0.021 | 0.019 |

**Table 5.** Precision, Recall, and $F_1$-Score Comparison of CNN, LeNet, and Deep CNN Models on the Pashto Character Dataset.

| Parameters | CNN | LeNet | Deep CNN |
|---|---|---|---|
| Precision | 93.5 | 95.5 | 95.7 |
| Recall | 93.5 | 95.7 | 95.7 |
| F-Score | 93.5 | 95.6 | 95.7 |

It can be seen from Table 4 that the accuracy for CNN was 98.7 percent, and the LeNet model was 91.9 percent on testing data in the final epoch. The proposed Deep CNN model outperformed the other two models as it can be seen from Table 4 that the accuracy of the Deep CNN model was 99.1 percent on the similar detest in the last epoch. The loss for all three comparison models showed that CNN had a loss of 0.031. Similarly, LeNet had a loss of 0.021. The proposed Deep CNN showed slightly better performance than the other two models as the Deep CNN had a 0.032 loss for the testing in the 20th epoch. The comparison of the used models for accuracy and loss can be seen in Figures 13 and 14.

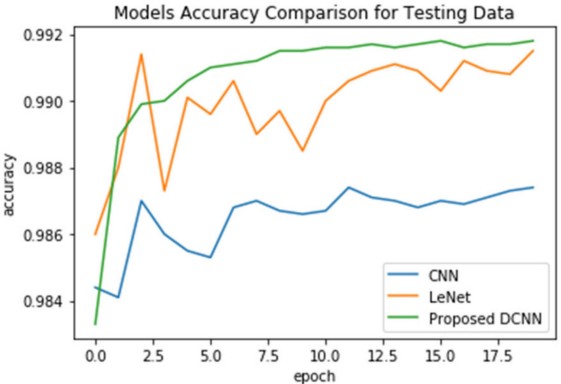

**Figure 13.** Accuracy Comparison of CNN, LeNet, and the Proposed DCNN on the Pashto Digits Dataset.

Similarly, Table 5 shows the precision, recall, and F1-Score comparison of the used models. Table 5 shows that CNN had a precision of 93.5 percent, a recall of 93.5 percent, and an F1-Score of 93.5 percent for testing data. Similarly, LeNet had a precision of 95.5 percent, a recall of 95.7 percent, and an F1-Score of 95.6 percent. The proposed Deep CNN model showed better performance than the other two models as it had a precision of 95.7 percent, a recall of 95.7 percent, and an F1-Score of 95.7 percent.

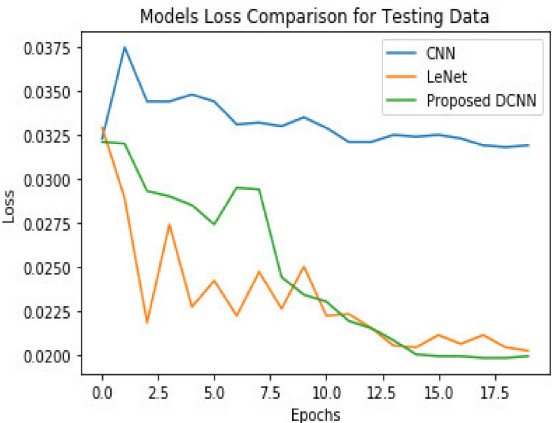

**Figure 14.** Loss Comparison of CNN, LeNet, and the Proposed DCNN on the Pashto Digits Dataset.

*4.5. Combined Pashto Digit and Character Dataset*

Pashto digits and characters datasets were combined to form an image dataset. The dataset consisting of 106,000 for 53 classes (43 for characters and 10 for digits) was divided into 80 percent for training and 20 percent for testing. The performance of the models was mainly analyzed using accuracy and loss. All the models showed less accuracy as compared to the previous two experiments, because when Pashto digits and characters are combined, usually, conflict occurs between many classes because the shape of Alif (ا) and One (١), wawo (و) and nine (٩), hey (ه) and five (٥), and ayen (ع) and four (٤) are somewhat the same. Due to this problem, the system cannot easily differentiate between these letters and hence performance is affected.

It can be seen from Table 6 that the accuracy for CNN was 66.5 percent for testing data in the final epoch. Similarly, the accuracy for LeNet was 69.8 percent for testing data. The proposed Deep CNN model outperformed the other two models as it can be seen from Table 5 that the accuracy of the Deep CNN model was 70.6 percent for testing data in the final 20th epoch. Similarly, the loss for CNN was 0.307, LeNet had a loss of 0.321, and the proposed Deep CNN had a better loss of 0.330 for testing data. Figures 15 and 16 show the accuracy and loss comparison of the three models.

**Table 6.** Accuracy and Loss Comparison of CNN, LeNet, and Deep CNN Models on the Combined Pashto Digit and Character Dataset.

| Accuracy/MSE | CNN | LeNet | Deep CNN |
|---|---|---|---|
| Accuracy | 66.5 | 69.8 | 70.6 |
| MSE | 0.307 | 0.031 | 0.303 |

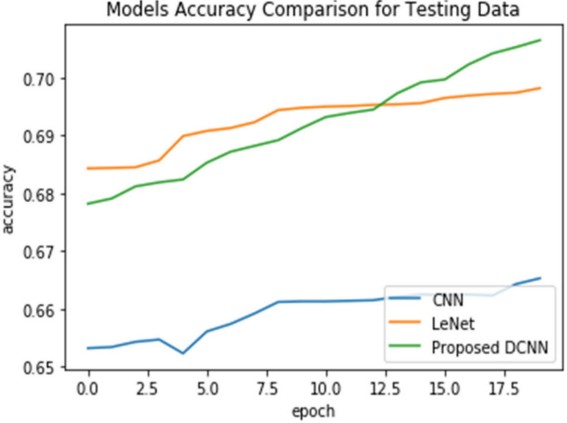

**Figure 15.** Accuracy Comparison of CNN, LeNet, and the Proposed DCNN on the Combined Pashto Digit and Character Dataset.

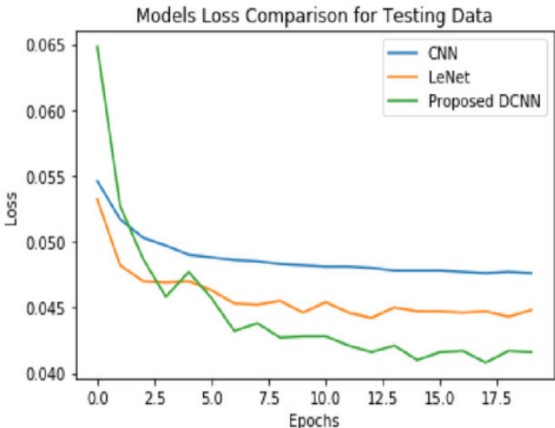

**Figure 16.** Loss Comparison of CNN, LeNet, and the Proposed DCNN on the Combined Pashto Digit and Character Dataset.

Similarly, Table 7 shows the precision, recall, and F-Score comparison of the used models. Table 7 shows that CNN had a precision of 49.1 percent, a recall of 48.7 percent, and an F1-Score of 48.7 percent for testing data. Similarly, LeNet had a precision of 60.2 percent, a recall of 53.6 percent, and an F1-Score 56.8 percent. The proposed Deep CNN model showed a better performance than the other two models as it had a precision of 62 percent, a recall of 53.9 percent, and an F1-Score of 57.7 percent.

**Table 7.** Precision, Recall, and $F_1$-Score Comparison of CNN, LeNet, and Deep CNN Models on the Combined Pashto Digit and Character Dataset.

| Parameters | CNN | LeNet | Deep CNN |
|---|---|---|---|
| Precision | 49.1 | 60.2 | 62.0 |
| Recall | 48.7 | 53.6 | 53.9 |
| F-Score | 48.7 | 56.8 | 57.7 |

Figure 17 shows the comparison of the three models in terms of these performance parameters.

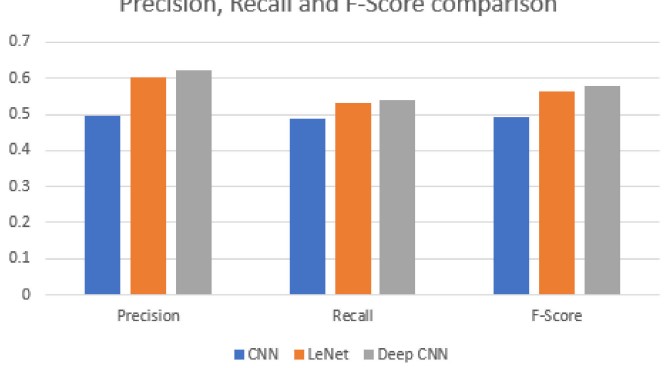

**Figure 17.** Precision, Recall, and $F_1$-Score Comparison of CNN, LeNet, and the Proposed DCNN on the Combined Pashto Digit and Character Dataset.

## 5. Conclusions and Future Work

Three CNN models with different set of layers were evaluated. The proposed trio Deep CNN model with deeper convolution provided the best result on testing data, in terms of loss (0.020 for Pashto characters, 0.019 for Pashto digits, and 0.303 for combined Pashto digits and characters) and accuracy (99.4 percent for Pashto characters, 99.1 percent for Pashto digits, and 70.6 percent for combined Pashto digits and characters datasets). As layers decreased, the accuracy for LeNet lowered to 99.2 percent for Pashto characters,

99.1 percent for Pashto digits, and 69.8 percent for combined Pashto digits and characters datasets.

The third model in the comparison, i.e., CNN, had an accuracy of 98.3 percent for Pashto characters, 98.7 percent for Pashto digits, and 66.5 percent for Combined Pashto digits and characters datasets. The F-measure of the proposed Deep CNN on test data (99.4 for Pashto characters, 95.7 for Pashto digits, and 57.7 for combined) was higher than LeNet (99.2 for Pashto characters, 95.6 for Pashto digits, and 56.6 for combined) and CNN (98.2 for Pashto characters, 93.1 for Pashto digits, and 56.5 for combined). The recall of Deep CNN (99.4 for character, 95.7 for digits, and 53.8 for combined) was higher than LeNet (99.2 for characters, 95.7 for digits, and 53.2 for combined) and CNN (98.2 for characters, 93.5 for digits, and 48.7 for combined). Similarly, the precision of Deep CNN (99.4 for characters, 95.7 for Pashto digits, and 62.2 for combined) was higher than LeNet (99.2 for characters, 95.5 for Pashto digits, and 60.2 for combined) and CNN (98.3 for Pashto characters, 93.5 for Pashto digits, and 49.5 for combined).

This research shows that increasing the number of layers in Convolutional Neural Network gives better results on datasets and decreasing them decreases the accuracy. All the models in this study have better performance than the previous study on Pashto characters on different networks performed by [26], which had 72 percent accuracy, thus making Deep CNN the current best model for Pashto OCR classification. In the future, the effect of other optimization functions such as the Adam Optimizer can be examined to test Pashto OCR performance. Furthermore, the proposed model can be used to test handwritten Pashto cursive script. Similarly, the proposed Deep CNN model can also be utilized for different handwriting styles. The same training model can be trained to recognize handwritten text in live videos.

**Author Contributions:** Conceptualization, M.Z.R., N.M.N. and A.K.; Formal analysis, M.Z.R., N.M.N., M.A. and A.K.; Funding acquisition, M.Z.R. and N.M.N.; Investigation, M.Z.R., M.A. and A.K.; Methodology, M.Z.R., N.M.N., M.A. and A.K.; Resources, N.M.N.; Software, M.Z.R. and M.A.; Supervision, N.M.N. and A.K.; Validation, M.Z.R. and A.K.; Writing—original draft, M.Z.R., M.A. and A.K.; Writing—review & editing, M.Z.R., N.M.N. and A.K. All authors have read and agreed to the published version of the manuscript.

**Funding:** This work is financially supported by the Research & Management Office (RMC), Universiti Tun Hussein Onn Malaysia (UTHM) and Ministry of Higher Education (MOHE), Malaysia under Tier-1 research grant vote number H938.

**Acknowledgments:** The Authors would like to thank the Research & Management Office (RMC), Universiti Tun Hussein Onn Malaysia (UTHM) for supporting this research under Tier-1 research grant vote number H938.

**Conflicts of Interest:** The authors declare no conflict of interest.

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
