# Peer review of "Recognition of Cursive Pashto Optical Digits and Characters with Trio Deep Learning Neural Network Models"

_electronics, doi:10.3390/electronics10202508_

Round 1
Reviewer 1 Report
As there is no publicly available dataset for handwritten Pashto digits and available before this study, there is no work performed on the recognition of Pashto handwritten digits and character combined. Authors proposed a dataset of Pashto handwritten digits consisting of 60000 images. The trio deep learning Convolutional Neural Network i.e., CNN, LeNet and Deep CNN were trained and tested with both Pashto handwritten characters and digits datasets. From this perspective this paper is some novely approach and value added.
My comments and suggestions for authors:
- The 3rd contribution mentioned in line 67: "Use performance parameters like accuracy, loss, precision, recall and f-measure". If you inevt new measure it is your contribution, but if you only aply existing ones it is not a contribution.
- Figure 1 should be placed after point III. line 140 not before.
- For the scientific discussion I find olny 1 sentence at the end. Would recomment to supplement with comparing to other research results as well.
- Additional comment on future research direction would be nice for a reader.
Author Response
Dear Reviewer,
Thanks for the timely review. Hopefully, our response will be found adequate.
Regards,
Authors

Reviewer 2 Report
The paper studies recognition of Pashto digits and/or characters using CNN models. The topic is interesting and the models achieve good results.
My comments on the current manuscript are below:
1. One of the main contributions of this paper is to create a Handwritten Pashto digits dataset. I strongly suggest making this dataset public.
2. The models can achieve high accuracy (~99%) with either Character dataset or Digits dataset. Why does the accuracy decrease significantly with the combined dataset of Character dataset and Digits dataset? The manuscript must analyze it deeply.
3. Regarding the recognition models, their novelty or advantages are not very clear. What is the difference between CNN and Deep CNN? Therefore, the scientific contributions of the current manuscript are weak. Besides, some details of the proposed model are not clear. For example, what are the number of filters and their sizes, what is the batch size?
4. DCNN or CDNN in the whole manuscript?
Author Response
Dear Reviewer,
Thanks for spending time reviewing our article. We hope that the corrections are found adequate this time.
Regards,
Authors

Reviewer 3 Report
Overall, the work is quite interesting with a decent related work section and a fair experimental setup.
However, there are some serious weaknesses and parts of the manuscript should be revised:
- First, the biggest problem concerns the presentation of the experimental results. There are lots of redundant performance scores that add almost zero contribution to the work and confuse the reader. In Tables I, II, III, IV, V, VI, VII, VIII, IX performance ratings for different training epochs are presented. This is overkill and makes no sense. We are mostly interested in the final (at the end of the training process) performance ratings. If authors want to highlight differences in how different models are trained, they should create a dedicated paragraph/section and present the evolution of (probably) different behaviors with figures, i.e. lines/curves.
- Second, the metric of "loss" should not be presented so thoroughly. Some comments in the text are usually enough. Only if there is strange behavior on an algorithm, it makes sense to demonstrate additional results (e.g. the "loss" metric).
- Third, training performance can be omitted and only ratings on the test may be demonstrated. Four decimal digits on performance ratings are too many. Two (or three) are sufficient.
In general, results should be more compact. In addition to this there are some more issues:
- In Section "A. Data preprocessing" there is a "Conversion to Binary" step. Why does a normalization process follow, as long as all pixels have the value of 0 or 1? In addition to this, what is the purpose of integer to double conversion? It seems redundant.
- Figures 1 and 2 lack visual consistency and do not supplement the text adequately. They should be flatter and present the data flow horizontally.
- There is no need of giving the definition of the performance metrics in paragraph "B. Performance Parameters".
- References list is poor.
After these issues are fixed, the article should undergo another review cycle.
Author Response
Dear Reviewer,
Thanks for the timely review. Hopefully, our response will be found adequate.
Sincerely,
Authors

Round 2
Reviewer 2 Report
All my comments are answered well. It can be accepted.
Author Response
Dear Reviewer,
We hope that you are doing well. We have done all the required changes as per your suggestion.
Sincerely,
The Authors
Reviewer 3 Report
Unfortunately, the authors addressed none of my remarks, while the response letter seems like a draft with lots of syntax/grammar errors.
To my judge, the presentation of the results must be improved and some remarks/issues should be resolved (i.e. this kind of normalization step should be described in a single sentence or not even be mentioned - it is pretty basic knowledge).
Author Response
Dear Reviewer,
I hope that this time you will accept the changes in the document.
Warm Regards,
Authors

Round 3
Reviewer 3 Report
The manuscript has been improved a lot. It will be ready for publication if some minor adjustments will be made. For instance:
- There are some grammar/syntax (language) issues. For example, the statement in line 205 "because convolutional neural network performs better..." would be better in plural "because convolutional neural networks perform better...". An overall language/typo polishing should be made. Another example in lines 283-284 "0.994% accuracy, while the CNN, LeNet model achieved 0.983%, 0.9925 accuracy". I think all scores in this sentence are not percentages...
- Figures should have consistency and optimal presentation. For example, Fig. 14 seems vertically stretched. Second "Fig. 15" should be "Fig. 18"?
- Figures that present bar plots (like Fig. 15) are somewhat redundant, because there are also the same results in tables. I would recommend keeping only one type of presentation of the results, but this is highly subjective.
- Processing steps "Conversion to Binary", "Integer to double", and "Image Normalization Process" should be merged into one, simple paragraph. There is no need to describe a normalization process when data is binary. This is only about scaling the "high" value from 255 to 1. The reader, when comes across the "Image Normalization Process" title expects continuous data and an important reason for executing normalization. In this case, the process is typically a normalization process but it is such a simple case that this kind of extensive "documentation" confuses the reader.
To summarize, I think that an overall polishing should be conducted to achieve the quality needed for a scientific article. I would hit "major revision" because there are several corrections that should be made, but I choose "minor" because the manuscript has improved significantly and all the remaining are easy to be fixed.
Author Response
Respected Reviewer,
We hope that you are doing well and this time you will find our corrections acceptable. Thank you so much for spending time on our document.
Regards,
The Authors
